# Efficient Active Learning for Gaussian Process Classification by Error Reduction

**Guang Zhao**[1], **Edward R. Dougherty**[1], **Byung-Jun Yoon**[1,3], **Francis J. Alexander**[3], **& Xiaoning Qian**[1,2,3]

guangzhao@tamu.edu, falexander@bnl.gov,
{edward,bjyoon,xqian}@ece.tamu.edu

[1]**Department of Electrical & Computer Engineering,**
[2]**Department of Computer Science & Engineering,**
Texas A&M University
College Station, TX 77843, USA

[3]**Computational Science Initiative,**
Brookhaven National Laborator
Upton, NY 11973, USA

## Abstract

Active learning sequentially selects the best instance for labeling by optimizing an acquisition function to enhance data/label efficiency. The selection can be either from a discrete instance set (pool-based scenario) or a continuous instance space (query synthesis scenario). In this work, we study both active learning scenarios for Gaussian Process Classification (GPC). The existing active learning strategies that maximize the Estimated Error Reduction (EER) aim at reducing the classification error after training with the new acquired instance in a one-step-look-ahead manner. The computation of EER-based acquisition functions is typically prohibitive as it requires retraining the GPC with every new query. Moreover, as the EER is not smooth, it can not be combined with gradient-based optimization techniques to efficiently explore the continuous instance space for query synthesis. To overcome these critical limitations, we develop computationally efficient algorithms for EER-based active learning with GPC. We derive the joint predictive distribution of label pairs as a one-dimensional integral, as a result of which the computation of the acquisition function avoids retraining the GPC for each query, remarkably reducing the computational overhead. We also derive the gradient chain rule to efficiently calculate the gradient of the acquisition function, which leads to the first query synthesis active learning algorithm implementing EER-based strategies. Our experiments clearly demonstrate the computational efficiency of the proposed algorithms. We also benchmark our algorithms on both synthetic and real-world datasets, which show superior performance in terms of sampling efficiency compared to the existing state-of-the-art algorithms.

## 1 Introduction

Compared to traditional passive learning with randomly sampled training instances, active learning aims at "optimally" querying instances for labeling to achieve label efficiency when training machine learning models, especially when labeling is difficult or costly. There are two fundamental scenarios of active learning discussed in the literature: query synthesis and pool-based sampling [14]. In query synthesis, the leaner can request labels for any instance generated from a continuous feature space while pool-based sampling selects the instance from a finite set. Query synthesis is more challenging due to the infinite search space and inherent higher label uncertainty. Recent research on query synthesis with deep generative models has shown promising potential [18, 11]. However, in

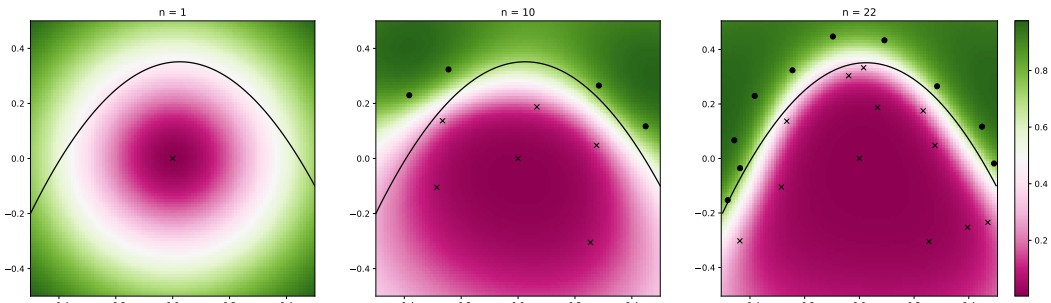

Figure 1: Phase diagram identification by active learning with Gaussian Process Classification (GPC).

many science and engineering applications, acquiring the label for even one instance is prohibitively resource-demanding and therefore active learning with deep models may not be practical.

For example, one of critical materials science research questions is to identify phase transition diagrams, where the phase transition response surface can be complex [8]. Identifying phase transitions can be formulated as finding the optimal classification boundaries between different phases in the enormous materials design space. However, precisely knowing the phase of each design with the corresponding input features requires costly and time-consuming materials synthesis and profiling experiments or running complex simulation models. Furthermore, there may exist significant uncertainty in acquired phase labels due to technical limitations. Hence, active learning for optimal Bayesian classifiers is a natural solution to help identify the phase diagram with as few as possible synthesized or simulated materials under such uncertainty and complexity. Gaussian Process Classification (GPC) as a popular and powerful Bayesian classifier with the flexibility of adopting different kernels [15], is suitable for solving this problem with appropriate active learning strategies. Fig.1 shows an example of phase identification in a two-dimensional design space by active learning with GPC (Details in the Appendix). In the figure, the black solid line indicates the phase transition response surface, the crosses and dots indicate the queries of different phases, and the colorbar indicates the predictive distribution of GPC. From the figure we can see, the predictive distribution identifies the phase transition boundary with a few samples guided by active learning using GPC surrogate models, which has significant cost and time savings compared to the traditional trial-and-error phase diagram identification paradigm in the current materials science practice.

Many active learning strategies have been proposed for GPC. For example, Bayesian Active Learning by Disagreement (BALD) selects the instance with the maximum mutual information between the observation and the derived uncertain model [5]. There also have been strategies targeting at reducing classification error directly or indirectly. Estimated Error Reduction (EER) strategies optimize for the error reduction after training with the new queries [13, 10, 6]. We note here that EER-based active learning has been studied for both Gaussian Process Regression (GPR) and GPC [13, 6]. For GPR, the regression error can be represented by the posterior predictive variance with the analytical expression and the acquisition function is easy to calculate for efficient active learning. In contrast, for GPC, there is no analytical expression for the posterior predictive. The model updates need approximate inference, such as Expectation Propagation (EP), an algorithm iteratively approximating the GPC posterior [7]. Moreover, the classification error computation requires the new query labels; so the calculation of the corresponding EER-based acquisition function requires incrementally retraining the model for each possible query label to calculate the expected future error as the utility to guide active learning. The complexity of training GPC with EP approximation is $O(n^3)$, where $n$ is the number of observed data. This has been the main hurdle to develop active learning for GPC models.

To reduce the calculation cost of EER, the paper [6] proposed to use a non-iterative but less accurate method Assumed Density Filtering (ADF) approximating the new posterior when calculating the acquisition functions. The paper [3] proposed a novel approximated error reduction (AER) criterion, in which the error reduction of a candidate is estimated based on the impact over its nearby instances. The approximated estimation avoids re-inferring the labels of massive instances. These methods are relatively efficient in computation. But besides EP approximation, they need additional approximations in calculating acquisition functions so the acquisition functions are not precisely calculated, which may degrade the desired data efficiency.

In this paper, within the EER-based active learning framework for GPC, we develop computationally efficient algorithms to compute EER to guide both query synthesis and pool-based active learning. In particular, we consider EER as the reduction of the Mean Objective Cost of Uncertainty (MOCU) [16] since the learning objective of GPC, in particular for identifying phase diagram, is to reduce the classification error. By deriving the joint distribution of queries as a one-dimensional integral, we avoid retraining the GPC for each query when calculating the EER/MOCU-based acquisition function. We further leverage a smooth approximation of MOCU, Soft MOCU (SMOCU) [17], to enable efficient gradient computation of the SMOCU reduction by deriving the corresponding chain rule for efficient query synthesis with GPC. To the best of our knowledge, this is the first algorithm for query synthesis active learning based on EER strategies. We show in our experiments that our algorithms accelerate the computation of the acquisition functions. Compared with other existing algorithms, we demonstrate consistent data efficiency of our algorithms with both synthetic and real-world datasets.

## 2 Problem Setting and Background

### 2.1 Gaussian Process Classification (GPC)

Consider a binary classification problem with the instance space $\mathcal{X}$ and binary label set $\mathcal{Y} = \{0, 1\}$, we aim to train a classifier $\psi : \mathcal{X} \to \mathcal{Y}$ to predict labels for unobserved instances $\psi(\boldsymbol{x}_*), \boldsymbol{x}_* \in \mathcal{X}$. The Gaussian Process classification (GPC) framework connects the instance and the label by a latent function $f$, which is a random process depending on $\boldsymbol{x}$. Assume that $f$ follows a Gaussian Process (GP) prior $f \sim \text{GP}(\mu(\cdot), k(\cdot, \cdot))$, where $\mu(\cdot)$ is a mean function and $k(\cdot, \cdot)$ is a covariance kernel function [9]. GP is a popular and powerful model for both regression and classification. A good property of GP is that given any finite number of instances $\boldsymbol{x}_i$, the joint distribution of $f(\boldsymbol{x}_i)$ is still Gaussian. For classification problems, given $f$, the label $y$ takes a Bernoulli distribution with probability $p(y = 1|\boldsymbol{x}, f) = \Phi(f(\boldsymbol{x}))$, where $\Phi$ is the Gaussian cumulative distribution function.

Given a sequence of observations $D = \{X, Y\}$ with $X = \{\boldsymbol{x}_1, \boldsymbol{x}_2, \ldots, \boldsymbol{x}_n\}$ and $Y = \{y_1, y_2, \ldots, y_n\}$, the class labels are conditionally independent given the latent function. Therefore, the joint likelihood can be factorized as: $p(Y|X, f) = \prod_i p(y_i|\boldsymbol{x}_i, f)$. Since the likelihood probability is non-Gaussian, the posterior $\pi(f|X, Y) \propto \pi(f)p(Y|f, X)$ can not be computed analytically and approximation are often adopted. The Expectation Propagation (EP) algorithm approximates $\pi(f|X, Y)$ with a Gaussian approximation $q(f|X, Y) = \mathcal{N}(f|\tilde{\mu}(\cdot), \tilde{\Sigma}(\cdot, \cdot))$ by iteratively moment matching marginal posteriors [9].

With the approximated posterior, the marginal distribution of $f_* = f(\boldsymbol{x}_*)$ is still Gaussian. Denote the mean and variance as $\mu_*$ and $\sigma_{**}$ respectively, then there is a closed-form expression for $p(y_*|\boldsymbol{x}_*, X, Y)$ as:

$$p(y_* = 1|X, Y, \boldsymbol{x}_*) = \int \Phi(f_*)\phi(f_*|\mu_*, \sigma_*^2)df_* = \Phi(\frac{\mu_*}{\sqrt{1 + \sigma_{**}}}). \tag{1}$$

Given the predictive probability, we assume the prediction of the instance is the most probable label $\arg\max_{y_*} p(y_*|X, Y, \boldsymbol{x}_*)$, which is known as the Optimal Bayesian Classifier (OBC) [1].

### 2.2 Bayesian Active Learning Based on Estimated Error Reduction (EER)

In the Bayesian active learning procedure, we consider a general case that the label given by the annotator is not deterministic, but the label follows an unknown distribution instead. Given the GPC model and observations, active learning sequentially chooses a query from a discrete instance set or a continuous instance space, to efficiently improve the prediction performance of GPC. In each iteration, the query is chosen by optimizing an acquisition function. The acquisition function reflects the anticipated benefit of the queries, and it usually depends on the model posterior $\pi(f|X, Y)$. After observing the new data $(\boldsymbol{x}_*, y_*)$, the model is updated by Bayes' rule to get the posterior $\pi(f|X, Y, \boldsymbol{x}_*, y_*)$. In the following content, we'll discuss the iteration of active learning after observing training data $(X, Y)$, for the sake of clarity, we omit $X, Y$ from the notations of posterior and posterior predictive distributions as $\pi(f)$ and $p(y|\boldsymbol{x})$.

Assume the distribution of instances over $\mathcal{X}$ is $\boldsymbol{x}_s \sim p(\boldsymbol{x}_s)$, denote the optimal Bayesian classifier (OBC) as $\psi_{\pi(f)}(\cdot)$. Based on GPC, the expected classification error of the OBC is:

$$\mathbb{E}_{\boldsymbol{x}_s}\{1 - p(y_s = \psi_{\pi(f)}(\boldsymbol{x}_s)|\boldsymbol{x}_s)\} = \mathbb{E}_{\boldsymbol{x}_s}\{1 - \max_{y_s} p(y_s|\boldsymbol{x}_s)\}. \tag{2}$$

In the GPC setting, if the latent function $f$ is known, we can accordingly label each instance with the most probable label as $\psi_f(\boldsymbol{x}) = \arg\max_y p(y|\boldsymbol{x}, f)$, where $\psi_f(\cdot)$ denotes the optimal classifier. The classification error of $\psi_f$ is $\mathbb{E}_{\boldsymbol{x}_s}\{1 - \max_y p(y|\boldsymbol{x}, f)\}$. Due to the model uncertainty, the classification error of OBC should be larger than that of the optimal classifier. The Mean Objective Cost of Uncertainty (MOCU) measures the classification error increase due to the model uncertainty [16]. It is defined as the OBC classification error minus the expected classification error of the optimal classifier under model uncertainty:

$$\mathcal{M}(\pi(f)) = \mathbb{E}_{\boldsymbol{x}_s}\{1 - \max_{y_s} p(y_s|\boldsymbol{x}_s)\} - \mathbb{E}_{\pi(f)}\{\mathbb{E}_{\boldsymbol{x}_s}[1 - \max_{y_s} p(y_s|\boldsymbol{x}_s, f)]\}. \tag{3}$$

As MOCU captures the direct influence of the model uncertainty on the classification error, we can take the MOCU reduction in a one-step-look-ahead setting as the acquisition function:

$$U^{\mathrm{M}}(\boldsymbol{x}_*; \pi(f)) = \mathcal{M}(\pi(f)) - \mathbb{E}_{y_*|\boldsymbol{x}_*}[\mathcal{M}(\pi(f|\boldsymbol{x}_*, y_*))] \tag{4}$$

The second term is the MOCU after observing $(\boldsymbol{x}_*, y_*)$, and averages over $p(y_*|\boldsymbol{x}_*)$ since we do not know the label of $y_*$ before observation. It is easy to verify that in (4), the $\max_{y_s} p(y_s|\boldsymbol{x}_s, f)$ terms in $\mathcal{M}(\pi(f))$ and $\mathcal{M}(\pi(f|\boldsymbol{x}_*, y_*))$ are cancelled and the acquisition function can be proved to be equivalent to the Expected Error Reduction (EER) of OBC [10, 17]:

$$\begin{aligned} U^{\mathrm{M}}(\boldsymbol{x}_*) &= \mathbb{E}_{\boldsymbol{x}_s}\{1 - \max_{y_s} p(y|\boldsymbol{x}_s)\} - \mathbb{E}_{y_*|\boldsymbol{x}_*}\{\mathbb{E}_{\boldsymbol{x}_s}[1 - \max_{y_s} p(y_s|\boldsymbol{x}_s, \boldsymbol{x}_*, y_*)]\} \\ &= \mathbb{E}_{\boldsymbol{x}_s}\{\mathbb{E}_{y_*|\boldsymbol{x}_*}[\max_{y_s} p(y_s|\boldsymbol{x}_s, \boldsymbol{x}, y)] - \max_{y_s} p(y_s|\boldsymbol{x}_s)\}, \end{aligned} \tag{5}$$

Although MOCU reduction or EER is optimal for single queries, with noisy observation labels, it may get stuck before converging to the optimal classifier as discussed in [16]. To address this issue, the paper [17] proposed a smooth concave approximation of MOCU, called Soft-MOCU (SMOCU), which replaces the $\max_{y_s} p(y_s|\boldsymbol{x}_s)$ term in (3) with a LogSumExp term:

$$\mathcal{M}^{\mathrm{S}}(\pi(f)) = \mathbb{E}_{\boldsymbol{x}_s}\{1 - \frac{1}{k}\mathtt{LogSumExp}(k \cdot p(y|\boldsymbol{x}_s)\} - \mathbb{E}_{\pi(f)}\{\mathbb{E}_{\boldsymbol{x}_s}[1 - \max_{y_s} p(y_s|\boldsymbol{x}_s, f)]\}, \tag{6}$$

where $k$ is a parameter controlling the approximation to MOCU. The resulting acquisition function based on the SMOCU reduction can be defined as:

$$U^{\mathrm{S}}(\boldsymbol{x}_*) = \mathbb{E}_{\boldsymbol{x}_s}\{\mathbb{E}_{y_*|\boldsymbol{x}_*}[\frac{1}{k}\mathtt{LogSumExp}(k \cdot p(y_s|\boldsymbol{x}_s, \boldsymbol{x}_*, y_*))] - \frac{1}{k}\mathtt{LogSumExp}(k \cdot p(y_s|\boldsymbol{x}_s))\}, \tag{7}$$

which guarantees the convergence to the optimal classifier [17]. The resulting acquisition function is a smooth function of $p(y_s|\boldsymbol{x}_s, \boldsymbol{x}, y)$ and has no maximization operators. We will leverage this smooth acquisition function to derive the first efficient gradient-based query synthesis active learning algorithm for GPC.

## 3 Efficient Active Learning for GPC

In this section, we present our EER-based active learning algorithms for GPC based on the acquisition functions defined by the MOCU and SMOCU reduction. At each iteration, with the updated GPC given previous observations, the acquisition function (5) or (7) can be optimized to guide the selection of the next query for active learning. We first present a straightforward algorithm for both the discrete instance set (pool-based sampling) and continuous instance space (query synthesis) scenarios, where the acquisition function is optimized by random optimization. Random optimization first collects a random sample set $\mathcal{X}_* \subset \mathcal{X}$ of size $M_1$, calculates the acquisition function for each sample in the set and then takes the instance with the maximum acquisition function value as the query.

To calculate (5) or (7), the integral over $\mathcal{X}$ space is not analytical. Hence we need to calculate the integral by Monte Carlo sampling with $M_2$ samples of $\boldsymbol{x}_s \in \mathcal{X}$. Define $g^{\mathrm{M}}(\boldsymbol{x}_s; \boldsymbol{x}_*)$ and $g^{\mathrm{S}}(\boldsymbol{x}_s; \boldsymbol{x}_*)$ such that $U^{\mathrm{M}}(\boldsymbol{x}_*) = \mathbb{E}_{\boldsymbol{x}_s}\{g^{\mathrm{M}}(\boldsymbol{x}_s; \boldsymbol{x}_*)\}$ and $U^{\mathrm{S}}(\boldsymbol{x}_*) = \mathbb{E}_{\boldsymbol{x}_s}\{g^{\mathrm{S}}(\boldsymbol{x}_s; \boldsymbol{x}_*)\}$. Let $g(\boldsymbol{x}_s; \boldsymbol{x}_*)$ denote either $g^{\mathrm{M}}(\boldsymbol{x}_s; \boldsymbol{x}_*)$ or $g^{\mathrm{S}}(\boldsymbol{x}_s; \boldsymbol{x}_*)$. For each $\boldsymbol{x}_s$, the calculation of $g(\boldsymbol{x}_s; \boldsymbol{x}_*)$ requires deriving the probability distribution of $p(y_s|\boldsymbol{x}_s)$ and $p(y_s|\boldsymbol{x}_s, \boldsymbol{x}_*, y_*), \forall y_* \in \mathcal{Y}$. Here, $p(y_s|\boldsymbol{x}_s)$ can be calculated directly from (1) while updating $p(y_s|\boldsymbol{x}_s, \boldsymbol{x}_*, y_*)$ needs incremental training of GPC with observations $\{X, Y, \boldsymbol{x}_*, y_*\}$ based on the EP approximation, and then calculating (1). The whole procedure of optimizing the acquisition function at the $n$-th iteration need to retrain GPC for each

possible pair $(\boldsymbol{x}_*, y_*)$, therefore, the EP approximation needs to be performed $2 \times M_1$ times. The whole procedure is illustrated as the pseudocode *Algorithm 1* in the Appendix.

There are three issues of the acquisition function calculation in the straight forward algorithm. First, we need a large number of samples $M_2$ to have a reliable estimation of the integral in (5) or (7). Second, the calculation of $p(y_s|\boldsymbol{x}_s, \boldsymbol{x}_*, y_*)$ requires incremental retraining of the GPC model for each pair of $(\boldsymbol{x}_*, y_*)$, with computational complexity $O(M_1 n^3)$. Third, even though $U^{\mathrm{S}}(\boldsymbol{x}_*)$ is a differentiable function of $p(y_s|\boldsymbol{x}_s, \boldsymbol{x}_*, y_*)$, in the algorithm we actually use the EP approximation $q(f|\boldsymbol{x}_*, y_*)$ to calculate $U^{\mathrm{S}}(\boldsymbol{x}_*)$, and it is impossible to calculate the gradient $\nabla q(f|\boldsymbol{x}_*, y_*)$ during the EP procedure.

We develop our EER-based active learning algorithms to address these three presented challenges: 1) By importance sampling leveraging inhere GPC assumptions, we reduce the required number of samples for estimating acquisition functions; 2) We compute $p(y_s|\boldsymbol{x}_s, \boldsymbol{x}_*, y_*)$ by deriving analytic solution to marginalize the joint distribution $p(y_s, y_*|\boldsymbol{x}_s, \boldsymbol{x}_*)$ from the approximated posterior $q(f|X, Y)$, which avoids the retraining with EP approximation; 3) More critically, we derive the first gradient-based query synthesis algorithm when using the SMOCU-based acquisition function by deriving the gradient $\nabla q(f|\boldsymbol{x}_*, y_*)$ for efficient active learning.

## 3.1   Importance Sampling

Regarding the issue in requiring the high sampling number of $\boldsymbol{x}_s$ for reliable estimation of the acquisition function, we notice that most of the kernels applied in GPC assume that the observed data only have influence on neighboring regions. Hence, we only need to account for the samples near $\boldsymbol{x}_*$ to estimate its influence on the classification error, thereafter the acquisition function for each new query. More specifically, we can use importance sampling to reduce the required sampling number. Reformulate the expectation $U(\boldsymbol{x}_*) = \mathbb{E}_{\boldsymbol{x}_s \sim p(\boldsymbol{x}_s)}[g(\boldsymbol{x}_s; \boldsymbol{x}_*)]$ over an assistant distribution $\tilde{p}(\boldsymbol{x}_s; \boldsymbol{x}_*)$ as: $U(\boldsymbol{x}_*) = \mathbb{E}_{\boldsymbol{x}_s \sim \tilde{p}(\boldsymbol{x}_s; \boldsymbol{x}_*)}[p(\boldsymbol{x}_s)g(\boldsymbol{x}_s; \boldsymbol{x}_*)/\tilde{p}(\boldsymbol{x}_s; \boldsymbol{x}_*)]$.

The variance of the expectation is minimized when $\tilde{p}(\boldsymbol{x}_s; \boldsymbol{x}_*)$ is proportional to $p(\boldsymbol{x}_s)g(\boldsymbol{x}_s; \boldsymbol{x}_*)$. But this requires knowledge of the value of $U(\boldsymbol{x}_*)$, which is the acquisition function that we try to estimate. In practice, we can choose $\tilde{p}(\boldsymbol{x}_s; \boldsymbol{x}_*) \propto k(\boldsymbol{x}_s, \boldsymbol{x}_*)p(\boldsymbol{x}_s)$ instead, as $k(\boldsymbol{x}_s, \boldsymbol{x}_*)$ reflects the non-zero region of $g(\boldsymbol{x}_s; \boldsymbol{x}_*)$: $k(\boldsymbol{x}_s, \boldsymbol{x}_*) \approx 0$ means $(\boldsymbol{x}_s, y_s)$ and $(\boldsymbol{x}_*, y_*)$ are independent, thus $p(y_s|\boldsymbol{x}_s, \boldsymbol{x}_*, y_*) \approx p(y_s|\boldsymbol{x}_s)$ and $g(\boldsymbol{x}_s; \boldsymbol{x}_*) \approx 0$. For example, if $p(\boldsymbol{x}_s)$ is uniformly distributed within a finite region, $k(\boldsymbol{x}_s, \boldsymbol{x}_*)$ is a square exponential kernel, then $\tilde{p}(\boldsymbol{x}_s; \boldsymbol{x}_*)$ can be chosen as a truncated Gaussian distribution. Note that importance sampling and random optimization is only suitable for continuous instance space, or discrete set with large cardinality, For small instance set, we can traverse all the elements for calculating the expectation and optimizing the acquisition function.

## 3.2   Joint Distribution Calculation

To avoid the retraining of GPC for each $(\boldsymbol{x}_*, y_*)$, we can calculate the posterior predictive by $p(y_s|\boldsymbol{x}_s, \boldsymbol{x}_*, y_*) = p(y_s, y_*|\boldsymbol{x}_s, \boldsymbol{x}_*)/p(y_*|\boldsymbol{x}_*)$, which requires computing the joint distribution of $p(y_s, y_*|\boldsymbol{x}_s, \boldsymbol{x}_*)$. We remind the reader all the probabilities are conditioned on $\{X, Y\}$ and we omit them for the seek of brevity. Denote $f_s = f(\boldsymbol{x}_s)$ and $f_* = f(\boldsymbol{x}_*)$. Now we show how to simplify the calculation of $p(y_s, y_*|\boldsymbol{x}_s, \boldsymbol{x}_*)$. In the Gaussian approximation of the posterior $q(f|X, Y)$, the joint distribution of $f_s, f_*$ is still Gaussian. Since $y_s$ and $y_*$ are conditionally independent given $f_s$ and $f_*$, the joint distribution can be expressed as:

$$p(y_s = 1, y_* = 1|\boldsymbol{x}_s, \boldsymbol{x}_*) = \iint \Phi(f_s)\Phi(f_*)\phi(f_s, f_*|\mu_{s*}, \Sigma_{s*})df_s df_*, \qquad (8)$$

where $\mu_{s*}$ and $\Sigma_{s*}$ are the marginal mean and covariance matrix of $f_s$ and $f_*$. This integral can be simplified as a one-dimensional integral. Denote $\mu_{s*} = \begin{pmatrix} \mu_s \\ \mu_* \end{pmatrix}$ and $\Sigma_{s*} = \begin{pmatrix} \sigma_{ss} & \sigma_{s*} \\ \sigma_{s*} & \sigma_{**} \end{pmatrix}$. We can decompose the joint Gaussian distribution as the marginal distribution of $f_s$ times the conditional distribution of $f_*$ given $f_s$, i.e. $\phi(f_s, f_*|\mu_{s*}, \Sigma_{s*}) = \phi(f_s|\mu_s, \sigma_{ss})\phi(f_*|\tilde{\mu}_*(f_s), \tilde{\sigma}_{**})$,

where $\tilde{\mu}_*(f_s) = \mu_* + (f_s - \mu_s)\sigma_{s*}/\sigma_{ss}$ and $\tilde{\sigma}_{**} = \sigma_{**} - \sigma_{s*}^2/\sigma_{ss}$. Then, (8) can be transformed as:

$$
\begin{aligned}
p(y_s = 1, y_* = 1|\boldsymbol{x}_s, \boldsymbol{x}_*) &= \iint \Phi(f_*)\Phi(f_s)\phi(f_s, f_*|\mu_{s*}, \Sigma_{s*})df_s df_*, \\
&= \iint \Phi(f_*)\phi(f_*|\tilde{\mu}_*(f_s), \tilde{\sigma}_{**})df_* \ \Phi(f_s)\phi(f_s|\mu_s, \sigma_{ss})df_s \\
&= \int \Phi(\frac{\tilde{\mu}_*(f_s)}{\sqrt{\tilde{\sigma}_{**} + 1}})\Phi(f_s)\phi(f_s|\mu_s, \sigma_{ss})df_s.
\end{aligned}
\tag{9}
$$

The last line is based on the integral equation introduced in [9]. The above equation (9) calculates the joint distribution with the 1-d integral in constant time. With the joint distribution $p(y_s = 1, y_* = 1|\boldsymbol{x}_s, \boldsymbol{x}_*)$, we can easily obtain the joint distribution of $y_s, y_*$ with other label pairs, and finally we can obtain the posterior predictive $p(y_s|\boldsymbol{x}_s, \boldsymbol{x}_*, y_*)$ without retraining the GPC with EP approximation. Based on this solution with importance sampling, we can develop efficient algorithms estimating MOCU/SMOCU reduction. Combining with Random Optimization, we develop the active learning algorithm with MOCU reduction named Non-Retraining MOCU reduction with Random Optimization (NR-MOCU-RO), and similarly for SMOCU reduction NR-SMOCU-RO. The pseudocode of NR-(S)MOCU-RO can be found in the Appendix (*Algorithm 2*).

### 3.3 Gradient Calculation

With the introduced marginalization strategy and importance sampling, we can significantly improve the computational efficiency for pool-based active learning with GPC. However, in the query synthesis problems, we would like to optimize the acquisition functions with gradient-based algorithms. Usually the acquisition functions are multi-modal in the feature space, so the common practice is to perform random optimization first, and then take the optimal point as the initial point to perform the gradient-based algorithms [4].

Here we consider the gradient calculation of the acquisition function based on the SMOCU reduction $\nabla U^S(\boldsymbol{x}_*)$ for query synthesis active learning as $U^S(\boldsymbol{x}_*)$ is a smooth function whose gradients exist everywhere. For a Gaussian Process, the gradients of its mean and covariance functions have closed-form expressions of the gradients of its adopted kernel function. Here we assume that given the EP approximation $q(f|X, Y)$ and any pair of points $(\boldsymbol{x}_s, \boldsymbol{x}_*)$, the gradients $\nabla\mu_*, \nabla\sigma_{**}$ and $\nabla\sigma_{s*}$ with respect to $\boldsymbol{x}_*$ are known. Calculation of these gradients is detailed in Appendix E. With this assumption, we can use the chain rule to finally express the gradients $U^S(\boldsymbol{x}_*)$ in the form of $\nabla\mu_*, \nabla\sigma_{**}$ and $\nabla\sigma_{s*}$.

In (7), the second term is unrelated to $\boldsymbol{x}_*$, so the gradient of the SMOCU reduction is:

$$
\begin{aligned}
\nabla U^S(\boldsymbol{x}_*) = \mathbb{E}_{\boldsymbol{x}_s}[\nabla g^S(\boldsymbol{x}_s, \boldsymbol{x}_*)] &= \nabla\mathbb{E}_{\boldsymbol{x}_s}\{\sum_{y_*} p(y_*|\boldsymbol{x}_*)\frac{1}{k}\texttt{LogSumExp}[k \cdot p(y_s|\boldsymbol{x}_s, \boldsymbol{x}_*, y_*)]\} \\
&= \sum_{y_*} \nabla p(y_*|\boldsymbol{x}_*) \cdot \mathbb{E}_{\boldsymbol{x}_s}\{\frac{1}{k}\texttt{LogSumExp}[k \cdot p(y_s|\boldsymbol{x}_s, \boldsymbol{x}_*, y_*)]\} \\
&+ \sum_{y_*} p(y_*|\boldsymbol{x}_*) \cdot \mathbb{E}_{\boldsymbol{x}_s}\{\nabla\frac{1}{k}\texttt{LogSumExp}[k \cdot p(y_s|\boldsymbol{x}_s, \boldsymbol{x}_*, y_*)]\}.
\end{aligned}
\tag{10}
$$

In the first term, $\nabla p(y_*|\boldsymbol{x}_*)$ can be calculated with $\nabla\mu_*$ and $\nabla\sigma_{**}$ based on (1). For the second term, we can use the chain rule to compute $\nabla\texttt{LogSumExp}[k \cdot p(y_s|\boldsymbol{x}_s, \boldsymbol{x}_*, y_*)] = g_1 \cdot g_2$, where:

$$
g_1 = \frac{\partial\texttt{LogSumExp}[k \cdot p(y_s|\boldsymbol{x}_s, \boldsymbol{x}_*, y_*)]}{\partial p(y_s|\boldsymbol{x}_s, \boldsymbol{x}_*, y_*)}, \qquad g_2 = \nabla p(y_s|\boldsymbol{x}_s, \boldsymbol{x}_*, y_*).
\tag{11}
$$

Since $p(y_s|\boldsymbol{x}_s, \boldsymbol{x}_*, y_*) = p(y_s, y_*|\boldsymbol{x}_s, \boldsymbol{x}_*)/p(y_*|\boldsymbol{x}_*)$ and we have calculated $\nabla p(y_*|\boldsymbol{x}_*)$, now we only need to calculate $\nabla p(y_s, y_*|\boldsymbol{x}_s, \boldsymbol{x}_*)$ based on (9):

$$
\begin{aligned}
\nabla p(y_s = 1, y_* = 1|\boldsymbol{x}_s, \boldsymbol{x}_*) &= \nabla \int \Phi(f_s)\phi(f_s|\mu_s, \sigma_{ss})\Phi(\frac{\tilde{\mu}_*(f_s)}{\sqrt{\tilde{\sigma}_{**} + 1}})df_s \\
&= \int \Phi(f_s)\phi(f_s|\mu_s, \sigma_{ss})\phi(\frac{\tilde{\mu}_*(f_s)}{\sqrt{\tilde{\sigma}_{**} + 1}}) \cdot \nabla(\frac{\tilde{\mu}_*(f_s)}{\sqrt{\tilde{\sigma}_{**} + 1}})df_s,
\end{aligned}
\tag{12}
$$

which is again a 1-d integral. The gradient of $\nabla(\frac{\tilde{\mu}_*(f_s)}{\sqrt{\tilde{\sigma}_{**}+1}})$ can be again calculated by chain rule, and connected to the calculation of $\nabla \tilde{\mu}_*(f_s)$ and $\nabla \tilde{\sigma}_{**}$:

$$\nabla \tilde{\mu}_*(f_s) = \nabla \mu_* + \frac{f_s - \mu_s}{\sigma_{ss}} \nabla \sigma_{s*}, \qquad \nabla \tilde{\sigma}_{**} = \nabla \sigma_{**} - \frac{2\nabla \sigma_{s*}}{\sigma_{ss}}. \tag{13}$$

Therefore, $\nabla(\frac{\tilde{\mu}_*(f_s)}{\sqrt{\tilde{\sigma}_{**}+1}})$ is a linear function of $f_s$, and we can use numerical integral methods to calculate (12). The query synthesis algorithm with the integral computation is summarized in the pseudocode: NR-SMOCU with Stochastic Gradient Descent (NR-SMOCU-SGD).

In summary, to reduce the number of samples $\boldsymbol{x}_s$ for reliable estimation of acquisition functions, our algorithm utilizes an importance sampling with an assistant distribution chosen according to the kernel function. By calculating the posterior predictive directly from the joint distribution, the algorithm avoids retraining GPC with EP approximation. The introduction of the joint distribution also enables the efficient calculation of the gradient of the smooth acquisition function, with which we develop an efficient active learning algorithm for query synthesis.

---

**NR-SMOCU-SGD:** n-th iteration

---

1: **function** GRADIENTOPT($p(\boldsymbol{x}), q(f|X,Y)$)
2:     Obtain initial point $\boldsymbol{x}_*$ from RANDOMOPT($p(\boldsymbol{x}), q(f|X,Y)$)
3:     **while** not converge **do**
4:         Sample $M_2$ samples of $\boldsymbol{x}_s \sim \tilde{p}(\boldsymbol{x}_s; \boldsymbol{x}_*)$
5:         Calculate $p(y_*|\boldsymbol{x}_*)$ and $\nabla p(y_*|\boldsymbol{x}_*)$ by (1)
6:         **for** each $\boldsymbol{x}_s$ **do**
7:             **for** $y$ in $\{0,1\}$ **do**
8:                 Calculate $p(y_s, y_*|\boldsymbol{x}_s, \boldsymbol{x}_*)$ , $\nabla p(y_s, y_*|\boldsymbol{x}_s, \boldsymbol{x}_*)$ and $p(y_s|\boldsymbol{x}_s)$ by (1, 9, 12)
9:             **end for**
10:             Calculate $\nabla g^{\text{S}}(\boldsymbol{x}_s, \boldsymbol{x}_*)$ by (10 - 12)
11:         **end for**
12:         $\nabla U^{\text{S}}(\boldsymbol{x}_*) = \frac{1}{M_2} \sum_{\boldsymbol{x}_s} p(\boldsymbol{x}_s) \nabla g^{\text{S}}(\boldsymbol{x}_s; \boldsymbol{x}_*) / \tilde{p}(\boldsymbol{x}_s; \boldsymbol{x}_*)$
13:         Update $\boldsymbol{x}_*$ with $\nabla U^{\text{S}}(\boldsymbol{x}_*)$
14:     **end while**
15:     **return** $\boldsymbol{x}_*$
16: **end function**

---

## 4 Experiments

In this section we demonstrate the efficiency of our active learning algorithms combined with either random optimization (NR-MOCU-RO, NR-SMOCU-RO) or Adagrad (NR-SMOCU-SGD) in the following sets of experiments. In the first set of experiments, we analyze and benchmark the running time of our algorithm by comparing to the naive computation of the MOCU/SMOCU reduction. Then we benchmark our algorithms with other active learning algorithms for both query synthesis on synthetic benchmark datasets, and pool-based active learning on real-world datasets. The competing algorithms include random sampling, Maximum Entropy Search (MES) [12], Bayesian Active Learning by Disagreement (BALD) [5]; several MOCU-based active learning algorithms are also included to better show the accuracy and computation efficiency of our proposed algorithms, including the original MOCU and SMOCU (OR-MOCU/OR-SMOCU) algorithms as described in the beginning of Section 3. In addition, an ADF-MOCU algorithm is also included for comparison, which uses Assumed Density Filtering (ADF) to retrain the GPC [6] and importance sampling (IS) to calculate the expectation. In our experiments, we use GP prior for $f$ with the squared-exponential kernels $k(\boldsymbol{x}, \boldsymbol{x}') = \gamma^2 \exp(-\|\boldsymbol{x} - \boldsymbol{x}'\|^2/l^2)$, where $\{\gamma, l\}$ are model hyperparameters. The label probability is modeled with the probit function as $p(y|\boldsymbol{x}, f) = \Phi(f(\boldsymbol{x}))$. The code for our experiments is made available at `https://github.com/QianLab/NR_SMOCU_SGD_GPC`.

### 4.1 Estimation Accuracy And Running Time Comparison

We first evaluate the effect of using the joint distribution integral in calculating the acquisition functions. We compare the estimation of $p^{(I)}(y_s|\boldsymbol{x}_s, \boldsymbol{x}_*, y_*)$ through the joint distribution Integral

Table 1: Running time (in seconds) and estimation accuracy.

| | Algorithm | n = 10 | n = 100 | n = 500 | n = 1000 |
|---|---|---|---|---|---|
| **Time (s)** | naive | 0.125 | 0.675 | 64.2 | 846 |
| | IS | 0.126 | 0.674 | 64.2 | 848 |
| | IS+JDI | 0.568 | 0.591 | 0.598 | 0.643 |
| **Relative Error** | naive | 0.149 | 0.207 | 0.195 | 0.27 |
| | IS | 0.013 | 0.029 | 0.027 | 0.026 |
| | IS+JDI | 0.013 | 0.032 | 0.027 | 0.022 |

(9), with $p^{(R)}(y_s|\boldsymbol{x}_s, \boldsymbol{x}_*, y_*)$ estimated by Retraining GPC with EP approximation. For this set of experiments, we generate the initial data points using a latent function $f$ sampled from the GP prior. The instance space $\mathcal{X} = [-4, 4]$, and the hyperparameters $\gamma^2 = 0.5$, $l^2 = 1$. We initially sample 100 data points to train GPC, then we compare the values $g^{(I)} = \texttt{LogSumExp}(k \cdot p^{(I)}(y_s|\boldsymbol{x}_s, \boldsymbol{x}, y))/k$ and $g^{(R)} = \texttt{LogSumExp}(k \cdot p^{(R)}(y_s|\boldsymbol{x}_s, \boldsymbol{x}, y))/k$, since these are related to the calculation of the SMOCU reduction. With 1000 pairs of $(\boldsymbol{x}_s, \boldsymbol{x}_*)$ randomly sampled from $\mathcal{X}$, we calculate the relative error as $|g^{(I)} - g^{(R)}|/g^{(R)}$. The average relative error is *1.8e-5* and the maximum relative error is *2.4e-3*. We also compare the values $g^{(I)} = \max_{y_s} p^{(I)}(y_s|\boldsymbol{x}_s, \boldsymbol{x}, y)$ and $g^{(R)} = \max_{y_s} p^{(R)}(y_s|\boldsymbol{x}_s, \boldsymbol{x}, y)$, which is used to calculate the MOCU reduction. The average relative error is *1.3e-5* and the maximum relative error is *2.2e-3*. These results validate that using (9) can provide accurate estimates of the acquisition functions.

Next, we compare the running time and accuracy of estimating acquisition functions by three algorithms: 1) a naive algorithm calculating $p(y_s|\boldsymbol{x}_s, \boldsymbol{x}_*, y_*)$ with GPC retraining (naive), 2) sampling $\boldsymbol{x}_s$ by Importance Sampling and calculating $p(y_s|\boldsymbol{x}_s, \boldsymbol{x}_*, y_*)$ with GPC retraining (IS), 3) sampling $\boldsymbol{x}_s$ by Importance Sampling and calculating $p(y_s|\boldsymbol{x}_s, \boldsymbol{x}_*, y_*)$ with the Joint Distribution Integral (IS+JDI). The algorithms are implemented in Python 3.7 on a personal computer with Intel i5-10400 2.9 GHz CPU and 16G RAM. We set the initial datasets of size $n = 10, 100, 500$ and 1000, respectively. Note that in all three algorithms, inferring the predictive distribution for each $\boldsymbol{x}_s$ also takes a considerable part of computations, especially when $n$ is small, so we separate out the computation of the predictive distributions and directly compare the running time between retraining GPC (in the naive and IS algorithm) and calculating the joint distribution integral (in the IS+JDI algorithm). For all three competing algorithms, we sample 1000 $\boldsymbol{x}_s$'s, and calculate $U^{\mathrm{S}}(\boldsymbol{x}_*)$. We benchmark them with a "ground truth" algorithm: calculating $U^{\mathrm{S}}(\boldsymbol{x}_*)$ by the naive algorithm with *1e6 $\boldsymbol{x}_s$* samples. We perform this comparison for 100 $\boldsymbol{x}_*$'s. The average running time for a single $\boldsymbol{x}_*$ and relative error are shown in Table 1. From the table, the relative error of the naive algorithm with 1000 samples is much larger than the other two as expected since the other two algorithms use importance sampling with smaller estimation variance. As $n$ increase, the running time of both naive and IS algorithms increases fast since GPC training has a complexity of $O(n^3)$, while the running time of IS+JDI does not change much because the joint distribution integral is calculated in almost constant time. The results show that importance sampling does not impose much extra computation load in active learning, while it significantly increases the estimation accuracy. When $n$ is large, the joint distribution integral is faster than retraining GPC with EP approximation.

## 4.2 Query Synthesis

We now benchmark our proposed algorithms with a challenging synthetic dataset *checkerboard* $4 \times 4$, similar as the one tested in [5], which emulates the setup of phase diagram identification in materials science. Fig. 2a illustrates the classification boundaries. We perform all the competing active learning algorithms on this dataset. MES, BALD,RO-MOCU, RO-SMOCU, ADF-MOCU, NR-MOCU-RO and NR-SMOCU-RO are all optimized by random optimization with $M_1 = 1000$. In NR-SMOCU-SGD, we first perform random optimization with $M_1 = 800$ and set the best point as the initial point for Adagrad so that NR-SMOCU-SGD has similar running time compared with NR-SMOCU-RO at their corresponding setups for fair comparison. Algorithm performance is measured in terms of the error regret defined as the OBC error at $n$-th iteration minus the optimal classifier error of the simulated ground truth. To show the influence of the observation error on the performance of different algorithms, we further assume that there is a constant flip error rate on the observing labels. We report the results with different error rates equal to 0 and 0.2. In the experiments, we initially draw 100 samples for labeling to estimate the hyperparameters $\{\gamma, l\}$. Then we perform different

algorithms to collect new data. We repeat the procedure for 150 runs and plot the average error regret with standard deviations in Figs. 2b-c, for the flip error rate = 0.2, the running time of each algorithm in each iteration is also shown in Fig. 2d. The algorithm performance with the flip error rate of 0.1 is shown in the Appendix.

We can observe from these figures that MES does not performs well, which is because MES tends to query the points close to the decision boundary, while this problem has multiple intertwined boundaries and MES cannot differentiate different boundaries. OR-MOCU and OR-SMOCU perform worse than our proposed NR-MOCU-RO and NR-SMOCU-RO with respect to both computation and sample efficiency. That is because when compared with IS, the naive Monte Carlo used in OR-MOCU/OR-SMOCU with GPC cannot accurately estimate the acquisition function values. On the other hand, the running time of OR-MOCU/OR-SMOCU increases much faster than NR-MOCU-RO/NR-SMOCU-RO since they retrain the model for every query samples. ADF-MOCU runs faster than our proposed algorithms, but it performs poorly as the ADF approximation is not accurate enough. We also observe that as the error rate increases, the difference between NR-MOCU and NR-SMOCU also increases, that is because as the error rate increases, MOCU reduction tends to ignore the long-term effect of a query, leading to the degraded long-term performance, also discussed in [16]. Among these algorithms, NR-SMOCU-SGD performs better than other proposed algorithms by leveraging the gradient information in the optimization procedure.

### 4.3 Pool-Based Active Learning with Real-World Datasets

We also compare algorithms on the UCI datasets [2] for pool-based active learning. NR-SMOCU-SGD is not included as it is designed to search the continuous space. For each dataset, we split it into training and testing datasets. We take the training dataset as the sampling pool for active learning, initially we randomly choose two samples from each class for labelling, and use them to estimate the GPC hyperparameters. Then we apply the competing active learning algorithms to sequentially select the query from the training dataset and estimate the OBC error with the testing dataset after each iteration. Details of the tested UCI datasets are provided in the Appendix.

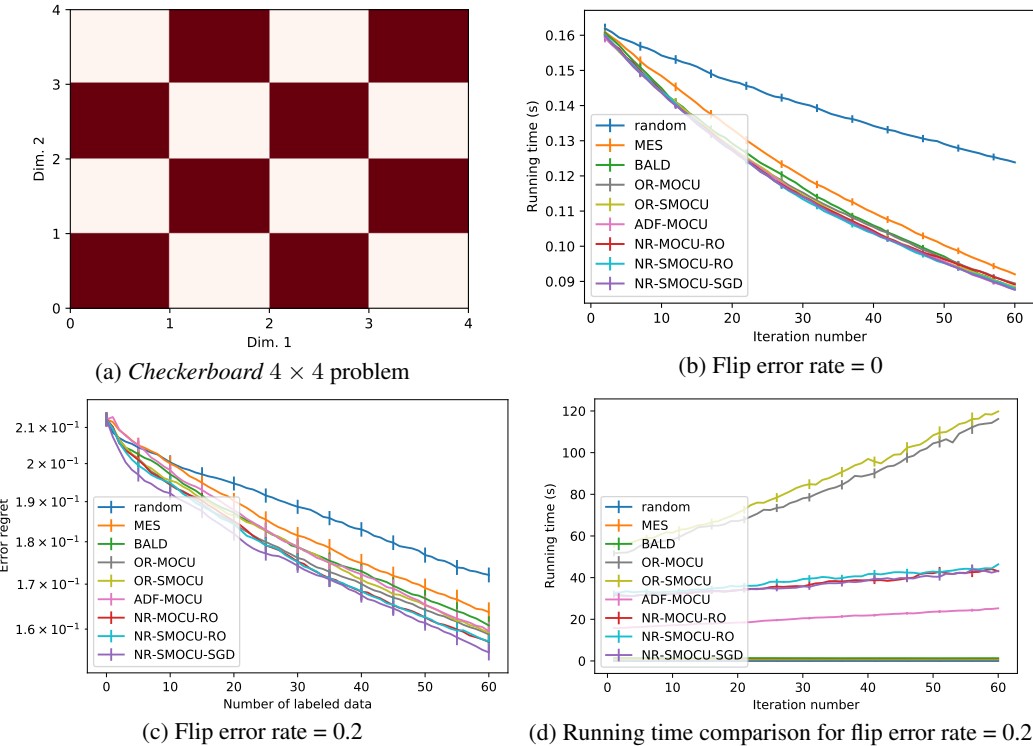

(a) *Checkerboard* $4 \times 4$ problem

(b) Flip error rate = 0

(c) Flip error rate = 0.2

(d) Running time comparison for flip error rate = 0.2

Figure 2: Comparison of expected OBC error regret and running time on the $4 \times 4$ *checkerboard* problem.

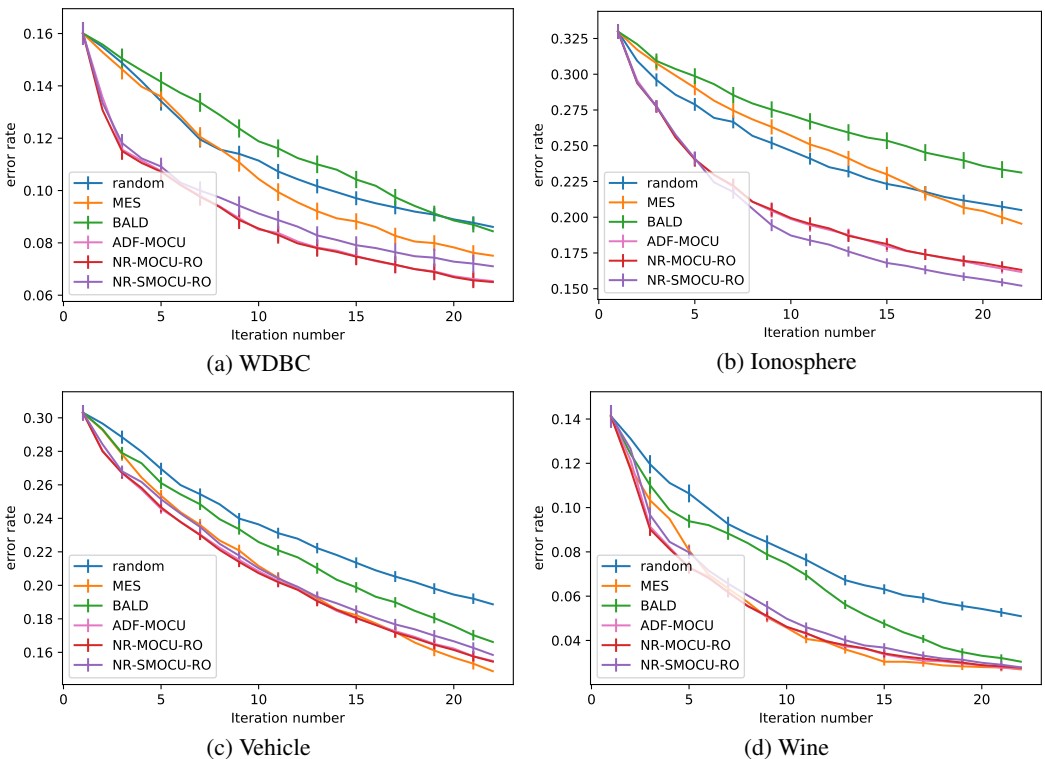

(a) WDBC

(b) Ionosphere

(c) Vehicle

(d) Wine

Figure 3: Classification error rate comparison on UCI datasets.

We repeat the active learning procedures for 100 runs and plot the average OBC error with standard deviation values for each algorithm for performance comparison shown in Fig. 3. In pool-based active learning, the expectation in MOCU/SMOCU is calculated by averaging over the whole instance pool, and there is no need for sampling, so the acquisition functions calculated by OR-MOCU/OR-SMOCU are almost the same as the ones calculated by NR-MOCU-RO/NR-SMOCU-RO. Therefore, performances of OR-MOCU/OR-SMOCU are not provided in the figure for the sake of conciseness. We notice that ADF-MOCU also performs similar to NR-MOCU-RO, though ADF provides rough approximations. That is because for the discrete space and small number of iterations, the small deviation due to approximation will not change the ranking of the instances in the pool. Overall, ADF-MOCU, NR-MOCU-RO and NR-SMOCU-RO are better than MES and BALD, which validates that MOCU-based algorithms achieve better sample, in particular label efficiency. Note that in the *Vehicle* and *Wine* datasets, MES performs closely to MOCU-based algorithms. When checking the converging GPC model, the classification boundary is relatively simple on these datasets with relatively low-dimensional feature space ($d \leq 18$). The *WDBC* and *Ionosphere* data are in higher dimensions ($d \geq 30$), for which our proposed algorithms perform significantly better.

## 5  Conclusions

We have proposed efficient EER-based active learning algorithms with GPC, which estimate the error reduction by querying instances based on the joint distribution of label pairs. We have derived the joint distribution as a one-dimensional integral with constant computation cost and calculate the predictive posterior based on it. Together with importance sampling, the acquisition function can be estimated efficiently by the 1-d integral of the joint distribution without incrementally retraining GPC with EP approximation, which has a computation complexity of $O(n^3)$. Without the need for EP approximation, we can further derive the chain rule to calculate the gradient of the SMOCU reduction, which provides us the first efficient gradient-based query synthesis active learning algorithm. Our experiments have demonstrated not only the accuracy and the running speed of our algorithms but also consistently better data/label efficiency compared with competing algorithms for both query synthesis and pool-based active learning.

**Acknowledgments** X. Qian was supported in part by the National Science Foundation (NSF) Awards 1553281, 1812641, 1835690, 1934904, and 2119103. B.-J. Yoon was supported in part by the NSF Award 1835690. The work of E. R. Dougherty and F. J. Alexander was supported by the U.S. Department of Energy, Office of Science, Office of Advanced Scientific Computing Research, Mathematical Multifaceted Integrated Capability Centers program under Award DE-SC0019303.

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
