# Efficient Active Learning for Gaussian Process Classification by Error Reduction: Appendix

**Guang Zhao**[1], **Edward R. Dougherty**[1], **Byung-Jun Yoon**[1,3], **Francis J. Alexander**[3], **& Xiaoning Qian**[1,2,3]

guangzhao@tamu.edu, falexander@bnl.gov,
{edward,bjyoon,xqian}@ece.tamu.edu

[1]**Department of Electrical & Computer Engineering,**
[2]**Department of Computer Science & Engineering,**
Texas A&M University
College Station, TX 77843, USA

[3]**Computational Science Initiative,**
Brookhaven National Laborator
Upton, NY 11973, USA

In this appendix, we provide more detailed descriptions of the phase identification example in the introduction section, the pseudocode of active learning algorithms, the tested datasets, as well as additional experimental results.

## 1 Details of the phase diagram active learning example in Introduction

The example in the introduction section of the main text is a synthetic binary phase diagram. The input features $\boldsymbol{x} = (x_1, x_2)$ belong to the domain $\mathcal{X} = [-0.5, 0.5]^2$ and the phase label $y$ is from the label set $\mathcal{Y} = \{0, 1\}$. The phase is decided by the phase boundary $f(\boldsymbol{x})$ as $y = \mathbb{1}(f(\boldsymbol{x}) > 0)$. In Fig. 1, the phase boundary is $f(\boldsymbol{x}) = x_2 - 2x_1^2 + 0.1x_1 + 0.35$. We apply our NR-MOCU-RO algoirthm to this problem. The active learning procedure has been shown in Fig. 1 in the main text, demonstrating the reliable phase identification with only 22 samples, corresponding to the number of required materials synthesis and profiling experiments in real-world materials science applications.

## 2 Additional pseudocode of active learning algorithms

We provide the pseudocode of the strait-forward EER-based active learning using random optimization in *Algorithm 1* and our NR-(S)MOCU-RO algorithm in *Algorithm 2*. Comparing two algorithms: *Algorithm 1* directly samples $\boldsymbol{x}_s \sim p(\boldsymbol{x})$ in the 3-rd line and retrains GPC for each $\boldsymbol{x}_*$ in the 7-th line, while *Algorithm 2* leverages importance sampling $\boldsymbol{x}_s \sim \tilde{p}(\boldsymbol{x}_s; \boldsymbol{x}_*)$ in the 5-th line and avoids retraining GPC by calculating the joint distribution $p(y_s, y_* | \boldsymbol{x}_s, \boldsymbol{x}_*)$ in the 8-th line.

## 3 Additional experiments and discussions

Here we test the algorithms in the task of finding the optimal classifier of the unknown probabilistic model $p(y|\boldsymbol{x}, f)$ with $f$ generated from the GP prior. In this set of experiments, the domain of $f$ is $\mathcal{X} = [-4, 4]$. Each $f$ is generated by first sampling 1000 function values from the GP prior with $\gamma^2 = 0.5$, $l^2 = 1$. $f$ is then given by the resulting GP posterior mean. We generate a total of 200 $f$'s following the procedure.

We perform all the competing active learning algorithms on these probabilistic models. MES, BALD, NR-MOCU-RO and NR-SMOCU-RO are all optimized by random optimization with $M_2 = 1000$. In NR-SMOCU-SGD, we first perform random optimization with $M_2 = 800$ and set the best point as the initial point for Adagrad so that NR-SMOCU-SGD has similar running time compared with NR-SMOCU-RO at their corresponding setups for fair comparison. Algorithm performance is measured in terms of the error regret defined as the OBC error at $n$-th iteration minus the optimal classifier

---
**Algorithm 1** Random EER-based active learning for GPC: n-th iteration
---
1: **function** RANDOMOPTIMIZATION($p(\boldsymbol{x}), q(f|X, Y)$)
2:     Sample $M_1$ samples of $\boldsymbol{x}_* \sim p(\boldsymbol{x})$
3:     Sample $M_2$ samples of $\boldsymbol{x}_s \sim p(\boldsymbol{x})$
4:     **for** each $\boldsymbol{x}_*$ **do**
5:         Calculate $p(y_*|\boldsymbol{x}_*)$ by (1)
6:         **for** $y_*$ in $\{0, 1\}$ **do**
7:             Use EP to approximate the posterior $q(f|\boldsymbol{x}_*, y_*)$
8:             **for** each $\boldsymbol{x}_s$ **do**
9:                 Calculate $p(y_s|\boldsymbol{x}_s)$ and $p(y_s|\boldsymbol{x}_s, \boldsymbol{x}_*, y_*)$ by (1)
10:                 Calculate $g(\boldsymbol{x}_s; \boldsymbol{x}_*)$
11:             **end for**
12:         **end for**
13:         $U(\boldsymbol{x}_*) = \frac{1}{M_2} \sum_{\boldsymbol{x}_s} g(\boldsymbol{x}_s; \boldsymbol{x}_*)$
14:     **end for**
15:     **return** $\tilde{\boldsymbol{x}} = \arg\max_{\boldsymbol{x}_*} U(\boldsymbol{x}_*)$
16: **end function**

---
**Algorithm 2** NR-(S)MOCU-RO: n-th iteration
---
1: **function** RANDOMOPTIMIZATION($p(\boldsymbol{x}), q(f|X, Y)$)
2:     Sample $M_1$ samples of $\boldsymbol{x}_* \sim p(\boldsymbol{x})$
3:     **for** each $\boldsymbol{x}_*$ **do**
4:         Calculate $p(y|\boldsymbol{x}_*)$ by (1)
5:         Sample $M_2$ samples of $\boldsymbol{x}_s \sim \tilde{p}(\boldsymbol{x}_s; \boldsymbol{x}_*)$
6:         **for** $y_*$ in $\{0, 1\}$ **do**
7:             **for** each $\boldsymbol{x}_s$ **do**
8:                 Calculate $p(y_s, y_*|\boldsymbol{x}_s, \boldsymbol{x}_*)$ by (9)
9:                 Calculate $p(y_s|\boldsymbol{x}_s)$ by (1)
10:                 Calculate posterior $p(y_s|\boldsymbol{x}_s, \boldsymbol{x}_*, y_*) = p(y_s, y_*|\boldsymbol{x}_s, \boldsymbol{x}_*)/p(y_s|\boldsymbol{x}_s)$
11:                 Calculate $g(\boldsymbol{x}_s; \boldsymbol{x}_*)$
12:             **end for**
13:         **end for**
14:         $U(\boldsymbol{x}_*) = \frac{1}{M_2} \sum_{\boldsymbol{x}_s} p(\boldsymbol{x}_s) g(\boldsymbol{x}_s; \boldsymbol{x}_*)/\tilde{p}(\boldsymbol{x}_s; \boldsymbol{x}_*)$
15:     **end for**
16:     **return** $\tilde{\boldsymbol{x}} = \arg\max_{\boldsymbol{x}_*} U(\boldsymbol{x}_*)$
17: **end function**

---

error of the simulated ground truth. Fig.S1 shows the average error regret with standard deviation bars in the logarithmic scale obtained by each algorithm across the 150 different probabilistic models. The results show that in the first few iterations, all three MOCU-based algorithms (NT-MOCU-RO, NT-SMOCU-RO, NT-SMOCU-SGD) outperform the competing algorithms. With more observations included, the decrease of the error regret slows down for NT-MOCU-RO. This is because the MOCU-based acquisition function cannot take into account the long-term effect of a query, also discussed in [3]. The plot also shows that the best algorithm in this setting is NR-SMOCU-SGD as it utilizes the gradient information during optimization.

We have also compared algorithms on the $4 \times 4$ *checkerboard* problem with flip error rate equal to 0.1 and the performances of algorithms are shown in Fig. S2. The figure shows similar trends as in Fig. 2b-c, which again demonstrate the superior performance of our proposed algorithms (NT-MOCU-RO, NT-SMOCU-RO, NT-SMOCU-SGD).

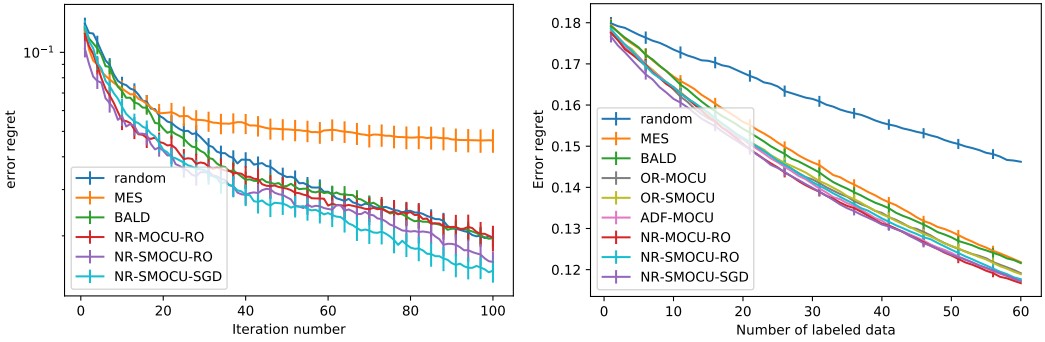

Figure S1: Algorithm performance comparison on 1-d GPC.

Figure S2: Algorithm performance comparison on the $4 \times 4$ *checkerboard* with flip error rate=0.1.

## 4 Data description

Detailed data characteristics of the UCI datasets [2] used in the main text and the experiments generating Fig. S2 are shown in Table S1.

Table S1: Details of the tested UCI datasets

| Dataset | $n_{\text{train}}$ | $n_{\text{test}}$ | $d$ | Dataset description |
|---------|--------|--------|-----|---------------------|
| WDBC | 284 | 285 | 30 | Wisconsin diagnostic breast cancer |
| Ionosphere | 175 | 176 | 34 | Radar returns from the ionospher |
| Vehicle | 208 | 208 | 18 | Features extracted from silhouettes image |
| Wine [1] | 65 | 65 | 13 | Wine quality |

## 5 Calculation of $\nabla \mu_*$, $\nabla \sigma_{**}$ and $\nabla \sigma_{s*}$

Given observations $X$ and $Y$, EP approximates the non-Gaussian likelihood $p(Y|X, f)$ with a Gaussian function: $p(Y|X, f) \approx C\mathcal{N}(f|\hat{\mu}, \hat{\Sigma})$. Here $C$ is a constant and $\hat{\mu}$ and $\hat{\Sigma}$ are calculated by iterative moment matching. Note that they are all unrelated to $x_*$. The posterior of $f$ is proportional to the likelihood times the prior by Bayes' rule and therefore is also a Gaussian distribution. Given a testing instance $x_*$, the posterior distribution of the latent variable $f_* = f(x_*) \sim \mathcal{N}(\mu_*, \sigma_{**})$. $\mu_*$ and $\sigma_{**}$ can be calculated based on the multiplication of two Gaussian functions:

$$\mu_* = \mathcal{K}(x_*, X)(\mathcal{K}(X, X) + \hat{\Sigma})^{-1}\hat{\mu}$$
$$\sigma_{**} = \mathcal{K}(x_*, x_*) - \mathcal{K}(x_*, X)(\mathcal{K}(X, X) + \hat{\Sigma})^{-1}\mathcal{K}(X, x_*),$$

where $\mathcal{K}$ is the kernel function. Since $\hat{\mu}$ and $\hat{\Sigma}$ are unrelated to $x_*$, the calculation of $\nabla \mu_*$ and $\nabla \sigma_{**}$ turns into calculating $\nabla \mathcal{K}(x_*, X)$ and $\nabla \mathcal{K}(x_*, x_*)$, which can be done as long as the kernel function is differentiable. Similarly, the mean and covariance matrix of the posterior joint distribution of $(f_*, f_s)$ can also be calculated by similar equations. Explicitly we have:

$$\sigma_{s*} = \mathcal{K}(x_s, x_*) - \mathcal{K}(x_s, X)(\mathcal{K}(X, X) + \hat{\Sigma})^{-1}\mathcal{K}(X, x_*),$$

and therefore we can calculate $\nabla \sigma_{s*}$. Given these gradients, we can calculate $\nabla_{x_*} p(y_s, y_* | x_s, x_*, X, Y)$ with equations (12) and (13).

## 6 The reason $\nabla_{x_*} p(y_s | x_s, x_*, y_* X, Y)$ is intractable when retraining GPC

Finally we explain why we cannot calculate the gradient $\nabla_{x_*} p(y_s | x_s, x_*, y_* X, Y)$ if we retrain the GPC with EP approximation for observations $X, Y, x_*, y_*$. We assume that EP approximates $p(Y, y_* | X, x_*, f)$ with the Gaussian function $C_* \mathcal{N}(f | \hat{\mu_*}, \hat{\Sigma_*})$. Now since $\hat{\mu_*}$ and $\hat{\Sigma_*}$ are related to $x_*$, we should also calculate the gradients of them. However, $\hat{\mu_*}$ and $\hat{\Sigma_*}$ are calculated numerically by iterative moment matching, so the gradients of them are intractable.