# OpenReview forum: "Efficient Active Learning for Gaussian Process Classification by Error Reduction"
_NeurIPS.cc/2021/Conference — NeurIPS 2021 Poster_

### Official Review · Reviewer_C1HE · 2021-07-14

**Rating:** 8
**Confidence:** 4

**Summary:**

this work derives an efficient way to compute the expected error reduction acquisition function for Gaussian Process classification active learning without requiring retraining under the well-known expectation propegation (EP) approximation for GP's. towards this end they use 1) importance sampling and 2) that the joint distribution of a training and test point of the Gaussian Process Classification model can be computed efficiently through a 1-dimensional integral. Their approach can also be used to derive gradients of the acquisition function which allows the approach to be used for query synthesis active learning. Furthermore, they empirically investigate the computational complexity and accuracy of their method, and benchmark the resulting active learning strategies.

**Ethical Concerns:**

nothing I can see

**Limitations And Societal Impact:**

nothing I can see

**Main Review:**

After author response: all my concerns have been addressed, and in view of the new experimental results, I have further increased my score.

I think this is a very well written, novel, and good paper and I really enjoyed reading it. It solves an important problem, making active learning with Gaussian Process classification much more feasible. My most pressing concern is that some obvious baselines [3,6,a] (for [a] see bottom) seem to be missing from the experiments. I would certainly raise my score further if these could be included in the experiments, irrespective of the actual outcome (e.g. if these baselines beat the proposed score, I would still vote for acceptance and further raise my score). Otherwise, there are some minor issues that should be further clarified regarding the experimental setup, and finally, since the paper's main contribution is computational efficiency I think this aspect should also be highlighted in the real world experiments.

Strengths
- well written, good quality
- novel and simple approach
- makes active learning with GP classification models computational feasible
- decent experiments

Weaknesses
- misses some obvious baselines [3,6,a] (I hope this can be resolved through openreview)
- computational efficiency should be reported in experiments for competing approaches
- there is no Discussion section discussing the weaknesses of the work (!!!) I really expect this to be addressed also during the review process...!
if the authors make a good effort to resolve these issues I would raise my score to 8 or 9.

Note: I didn't go through the mathematical derivation in detail, but it seems correct to me.

most important issues

1) Why are [3] and [6] not included in the benchmark? As they seem to be the most closely related to the current work.

2) I would really appreciate it if you would include a Gaussian Process regression model and it's corresponding acquisition strategy. My suspicion is that it could be very competitive with GP classification active learning; even though of course the model is misspecified. For example [a] (see at bottom) uses GP regression for active learning for a classification task. Their strategy is closely related to expected error reduction.

Note that; if the above baselines would be included I would for sure raise my score; irrespective of the outcome of the experiments. More comparisons only would improve the value of the work in my eyes, even if your strategy is outperformed by these other strategies. I think that the current work is so elegant and a nice solution to this problem, that even if the performance is not always competitive , it should definitely be accepted.

3) line 289: here you say that SMOCU performs better because MOCU has problems with long term effects. However, in Figure 2d I clearly see that SMOCU is performing worse than MOCU; but SMOCU-SGD _does_ perform better. Do you have any idea what is going on here? I think a remark is in order, the interpretation as I read it now is incorrect? Note that (Line 294) my intuition is that SGD may also have an advantage because it can more precisely pick locations of points than the other algorithms which are restricted to the finite pool. But it doesn't really

4) line 247: the relative absolute error seems small. however, I found in active learning the acquisition functions may sometimes only differ also by small values, thus small numerical inaccuracies can have a huge effect on what samples are selected. so to make this meaningful, I suggest you also give the minimal difference between two points their acquisition functions. this however, may be on the other hand too pessimistic, because the ordering might still be more or less the same. even more convincing that this will be no problem is to include the learning curve of the naive strategy without approximation versus the proposed strategies; these should overlap and have no significant differences if the approximation is truely good (which I suspect is the case). another option is to compute the ranking of all points in the pool for both methods and to compute the amount of ranking errors.

5) Since one of the main points of the work is computational efficiency, I think in the real world experiments it would be fair to also report on this aspect for all compared approaches.

medium importance

6) How was k determined for SMOCU? What is the value? I think I maybe missed this.

7) line 140-142: here definition of random optimization is confusing. is this now a baseline which just samples randomly, or is this an active learning strategy? it seems like the latter. I do not see why in lines 197 the active learning strategies are combined with random sampling... it seems to be almost part of the experimental setup. shouldn't there always be a pool that is sampled randomly? this seems like a default choice in pool-based AL which may not even need to be explicitly mentioned in the algorithm design, as all other approaches also use this component.

8) Figure 2 and 3: please put #training samples on the x-axis instead which is much clearer. Now it seems uncertain if iteration 1 corresponds to how many samples?

9) line 277: this must be M_1?

10) I would find it much more convincing if in Table 1 you would add an experiment with n = 500 or n = 1000, and illustrate that indeed the computation times blow up completely for naive and IS. if this is infeasible, perhaps these computation times can be estimated from a graph (e.g. plot computation time versus n, and fit a n^3 polynomial, and extrapolate the computation times). Now it is less convincing as the computation times are relatively still the same order of magnitude. By the way, is the computation time for the entire pool here?

11) line 284: how are the hyperparameters estimated; cross validation or using the marginal likelyhood? I suggest it may be wise to use cross validation to make a surfaceplot of the two hyperparameters, and have a look by hand if there is only a single optima or if there are multiple optima. then it may be better to hand pick one minima and fix the hyperparameters to that value, to reduce the variance of the experiments. this is not a problem if this is only done once, of course. I guess the data used to tune the hyperparameters is discarded afterwards? This wasn't entirely clear.

minor points

12) line 26: membership query framework is also a fundemental setup often studied, see Settles [14]
13) line 31-33: I would perhaps remove or change this claim regarding deep models, as this paper seems to pertain anyway to GPR models? But I agree: the material science motivation is good and keep that aspect; e.g. that the number of samples in such application is usually orders of magnitude smaller.
14) line 31: if you want to keep the deep learning link, here is a relevant paper on active learning with generative models:
Huijser, Miriam, and Jan C. van Gemert. "Active decision boundary annotation with deep generative models." Proceedings of the IEEE international conference on computer vision. 2017. which is also related to query synthesis.
15) line 130: what is stated here is not backed up by the equation which can be confusing. introduce the quantity, and keep the remark but relate it to [17].
16) line 143-147: here the mathematical notation and writing is really cumbersome and confusing. I think clearly eq 5 and 7 should define g^S and g^M, and then U^M and U^S should in turn be defined by just integrating out x_s! now the definitions are really terrible to get understand. your reader will also be happy if you clarify that the S stands for Smooth (still lost what the M stands for?)
17) line 154: "need a large number of samples M_2" to clarify that you are still talking about the M_2 samples
18) line 180: if the kernel is a square exponential, isn't then the proposal distribution just a Gaussian distribution? why the truncation?
19) section 4.1: was this experiment repeated multiple times with different f sampled from the GP prior?
line 261: maybe give the absolute error rate? I guess you mean here g=g' or p-p', or are you talking about the actual classification errors?

Suggestion for future work:
can we do the same trick to find 2 query points at the same time? Of course, computational complexity would be gigantic, but maybe SGD can make it work? This may be a better option to resolve the look ahead problem than SMOCU.

Note: if you run into pagelimit issues I would put the pseudo-code of the SGD alg. also in the appendix.

[a] Freytag, Alexander, Erik Rodner, and Joachim Denzler. "Selecting influential examples: Active learning with expected model output changes." European conference on computer vision. Springer, Cham, 2014.

**Time Spent Reviewing:**

3

---

> ### Author Response · Authors · 2021-08-10
> **Response to Reviewer C1HE**
>
> We truly appreciate the reviewer’s detailed and constructive comments. We here provide responses to the main critiques by the reviewer.
>
> Weakness of the work:
> We will provide more detailed discussions on limitations of the proposed work, including
> - As we have shown in the experiments, when $n$ is small, the computation speedup of the proposed strategies can be limited. However, this may not be critical as the proposed active learning is scalable with superior sample efficiency and has significant speedup when $n$ increases.
> -The derived joint distribution integral is only applicable to the probit likelihood function.
> -The acquisition function is defined as the expectation over the instance space, and we use Monte Carlo to calculate the expectation. In high-dimensional instance space (dimension larger than 100), finding a proper proposal distribution for importance sampling is difficult, so the calculation may require huge number of samples.
>
> If separate discussions on limitations and potential societal impacts are required, we will revise our final paper accordingly to better present these.
>
> **1-2**.  As the reviewer suggested, we’ll add more benchmark algorithms into our current comparison experiments and compare both the error rate and running time among these comparing algorithms. The benchmark algorithms include original MOCU/SMOCU and the methods in [3]. We will also try to include methods in [6, a], but these two algorithms have no open-source codes and may require more time to implement. Due to the limited time to post the rebuttal, we will report the numbers during the discussion period and add them into the final paper.
>
> **3**. We should be more careful in the performance analysis and will expand the discussion in the paper. If the function $f$ is sampled from a given prior, SMOCU would perform better than MOCU on average, as shown in the Figure S1 in the appendix of our original submission. Note in the checkerboard problem, the latent function $f$ is not continuous, and the kernel cannot correctly describe $f$, so it could be the case that MOCU outperforms SMOCU.
>
> We’d like to emphasize that in the experiments for Figure 2, the random sampled pools for the non-SGD active learning methods are different across iterations. Therefore, the non-SGD methods can also pick any locations from the instance space. But the selection does not use the information of the gradients and the optimization can be inefficient.
>
> **4**. We agree that the acquisition functions can also differ by small values, especially when $n$ is large. To show the effect of the small difference, we will include the naive MOCU/SMOCU algorithm into the comparison with the proposed algorithms.
>
> **5**. We will also include the running time of all compared approaches in our additional experiments.
>
> **6**. The optimal choice of $k$ can be affected by the total query budget and the specific classification problem. $k$ should be small if the sampling budget is large and/or the observation error rate is small; otherwise $k$ should be large. In our experiments, we just choose $k = 20$ for all the experiments.
>
> **7**. The reviewer is right that the random optimization is an optimization strategy. It’s used for optimizing acquisition functions without calculating the gradients. In this paper we use random optimization for comparison with the gradient-based optimization.
>
> **8**. We thank the reviewer for the helpful suggestion, we'll revise the figures accordingly
>
> **9**. The reviewer is correct that it should be $M_1$.
>
> **10**. As the reviewer suggested, we have run an additional experiment with $n$ up to 1000, in which we separate out the computation of the predictive distributions and directly compare the running time between retraining GPC (in the naive algorithm) and calculating the joint distribution integral (in the IS+JDI algorithm) when calculating the acquisition function $U^S(x_*)$. The running time with different $n$ is shown in the following table:
>
> |               | n=10| n=100|n = 500|n=1000|
> | ----------- | ----------- |------|---|-------|
> | retraining GPC| 0.125	|0.675|64.2|846|
> |  joint distribution integral| 0.568|0.591|0.598|0.643|
>
> It is obvious to see that with increasing $n$, the running time of retraining GPC increases in a cubic order as expected, while the joint distribution integral can be calculated in constant time. We will add these results with discussions and revise Table 1 in our final paper accordingly.
>
> **11**. The hyperparameters were estimated by maximizing the marginal likelihood. The data used to tune the hyperparameters is kept for the model training, which we believe is acceptable since the number of samples used for tuning is small (for example, in the UCI datasets, we only use four samples to tune the square exponential kernel with two hyperparameters.).
>
> **12-19**. We thank the reviewer for all these suggestions, we will update our paper accordingly.
>
> **16**. $M$ stands for MOCU since EER method is equivalent to the MOCU reduction method. We will make sure that all the acronyms in the final paper will be clearly defined.
>
> **18**. Here we assume that the uniform distribution is distributed over a finite interval.  Therefore, the Gaussian function times the uniform distribution is a truncated Gaussian distribution.
>
> **19**. The experiments do repeat multiple times with different $f$. The “absolute error rate” is a typo, we actually mean the absolute error ratio (or more common term “relative error”) of $U^S(x_*)$ computing by the three algorithms, comparing with the “ground truth”.
>
> Regarding the future work on batch active learning (querying two or more points in each iteration):
> As an acquisition function, EER or MOCU reduction may ignore the benefit of some queries, which is because the error rate only depends on the OBC [17]. Therefore, even the batch mode active learning based on EER may still ignore these queries and show myopic behaviors. Nevertheless, batch active learning is surely a significant direction to pursue as experiments may perform in parallel, and the main challenge is to solve the combinatorial optimization problem (subset selection) or adding reasonable diversity in the selected queries by heuristics.

---

### Official Review · Reviewer_U1Z1 · 2021-07-16

**Rating:** 5
**Confidence:** 3

**Summary:**

For the limitations of active learning in Gaussian Process Classification (GPC), the paper proposes an active learning algorithm based on EER with GPC, which shows computational efficiency and superior performance through experiments. Specifically, the paper derives the joint prediction distribution of queries as a one-dimensional integral to avoid retraining GPC for each query regarding the problems of active learning strategies in the GPC field, which reduces computational overhead. In addition, the paper also uses the Soft MOCU (SMOCU) to effectively calculate the gradient of the SMOCU reduction.

**Limitations And Societal Impact:**

Yes

**Main Review:**

This paper mainly considers the active learning scenario in GPC, which can be used to deal with the problem of identifying phase transitions in the field of materials science as the author says. Previous methods have some limitations, such as high computational overhead. Another problem is that EER cannot be applied to synthesis query with gradient optimization strategy.
This paper reduces the number of samples to estimate acquisition functions by importance sampling and proposes an appropriate solution to avoid retraining GPC for each query. The gradient based optimization method is applied to the synthesis query scenario by deriving $ \nabla q\left(f \mid \boldsymbol{x}_{*}, y_{*}\right) $.

However, the novelty of the paper is not enough. The improvement in computational efficiency is limited. The paper claims that the method proposed could avoid retraining GPC for each query, but it still needs much time to calculate the joint distribution integral. It can be seen from table 1 that when n=10, the IS+JDI method even costs more time than the naive method. The experiments are not solid. In the pool-based scenario, only one dataset is used. What's more, except for random, there are only two methods selected for comparison in the pool-based scenario, which is not convincing. In Section 3.3, this paper assumes that "given the EP approximation ... know the gradient ...", which is unrealistic. This paper is not well-written. For example, the formula such as `logsumexp` is not standard. Many formula and symbols are involved, but some of them are not clearly explained.


**Time Spent Reviewing:**

3

---

> ### Author Response · Authors · 2021-08-10
> **Response to Reviewer U1Z1**
>
> We thank the reviewer for the helpful comments. Here we provide the response to the comments by the reviewer:
>
> 1. Calculation of the joint distribution integral (JDI) requires more time when $n$ is small.
>
> The calculation of JDI takes the constant time comparing with retraining GPC, which requires the computation complexity $O(n^3)$. Therefore, although the calculation of JDI takes more time when $n$ is small, the calculation time with the IS+JDI method will not increase as $n$ increases; so it is scalable when $n$ is large. The additional experiment we have run can show the running time tendency in retraining GPC (in the naive algorithm) and calculating the joint distribution integral (in the IS+JDI algorithm):
>
>  |               | n=10| n=100|n = 500|n=1000|
> | ----------- | ----------- |------|---|-------|
> | retraining GPC| 0.125	|0.675|64.2|846|
> |  joint distribution integral| 0.568|0.591|0.598|0.643|
>
> We will add these results with discussions in our final paper accordingly.
>
> 2. Only two benchmark active learning algorithms MES and BALD are included in the comparison experiments.
>
> In Bayesian active learning, BALD is among the state-of-the-art methods [5] and MES (uncertainty sampling) is the simplest and most commonly used method. We will run additional experiments to include more benchmark algorithms into comparison, including original MOCU/SMOCU algorithm and the Approximated Error Reduction (AER) algorithm [3]. Due to the limited time to post the rebuttal, we will report the numbers during the discussion period and add them into the final paper.
>
>
> 3. In line 211, how do we calculate $\nabla \mu_*$, $\nabla \sigma_{**}$ and $\nabla \sigma_{s*}$ with respect to $x_*$ under the EP approximation?
>
> Given observations {$X, Y$}, EP approximates the non-Gaussian likelihood $ p(Y|X, f)$ with a Gaussian function as: $p(Y|X, f) \approx C \mathcal{N}(f|\hat{\mu}, \hat{\Sigma})$. Here $C$ is a constant and $\hat{\mu}$ and $\hat{\Sigma}$ are calculated by iterative moment match.
>      Given a testing instance $x_*$, the latent variable $f_* = f(x_*) \sim \mathcal{N}(\mu_*, \sigma_{**})$ where:
>      $ \mu_* = \mathcal{K}(x_*, X) (\mathcal{K}(X, X)+\hat{\Sigma})^{-1} \hat{\mu} $,
>
> $\sigma_{**} = \mathcal{K}(x_*, x_*) - \mathcal{K}(x_*, X)(\mathcal{K}(X, X)+\hat{\Sigma})^{-1}\mathcal{K}(X, x_*)$,
>
> where $\mathcal{K}$ is the kernel function.
>
> Since $\hat{\mu}$ and $\hat{\Sigma}$ are unrelated to $x_*$, the calculation of $\nabla \mu_* $ and $\nabla \sigma_{**}$ turn into the calculation of $\nabla \mathcal{K}(x_*, X)$ and $\nabla \mathcal{K}(x_*, x_*)$, which can be done as long as the kernel function is differentiable. Similarly, we can calculate $\nabla \sigma_{s*}$. We'll include all the details of the calculation in the appendix of the final paper.

---

### Official Review · Reviewer_gsQi · 2021-07-17

**Rating:** 5
**Confidence:** 4

**Summary:**

This paper proposes a bag of three techniques to improve the computational efficiency of  expected error reduction acquisition function for active learning Gaussian process classification models. The three techniques are
* importance sampling: theoretically the error reduction is integrated over the whole input space, which is intractable and Monte Carlo sampling is adopted. To reduce the number of Monte Carlo samples x_s, importance sampling is proposed to focus sampling nearby the query candidate x_star, which is achieved by sampling portional to the kernel function centered at x_star.
* avoid retraining for each query candidate by deriving the joint distribution of the query candidate label and Monte Carlo sample label, which become a one-dimensional integral.
* deriving the gradient of the acquisition function for the SMOCU acquisition function so that gradient-based optimization can be used in query synthesis setting.

Experiments
* simulation experiments are conducted to demonstrate the accuracy of the joint distribution method against naive retraining
* simulation experiments are conducted to demonstrate the accuracy and efficiency of the proposed method against naive computation of the acquisition function
* experiment on the checkerboard dataset to show the effectiveness of the proposed algorithms in query synthesis setting against baselines such as random, max entropy search (MES) and Bayesian active learning by disagreement (BALD), where 0, 10%, or 20% noise is added in observations.
* experiments on three UCI datasets to demonstrate the effectiveness of the proposed algorithm in pool-based active learning setting against MES and BALD.


**Main Review:**

This paper proposes a combination of three techniques to improve the efficiency of some recently proposed active learning policy. While all the three techniques are well motivated and seem mathematically solid, the contribution is somewhat incremental, and empirical results are not super convincing.

The main argument is improved efficiency but the only experiment to demonstrate this is on a 1d toy example where the advantage is not even that big, at least not presented in a convincing way.

I would also expect more ablation studies on the number of samples on more datasets. E.g., how does the error rate drop as we increase the number of samples, comparing random vs importance sampling.

The paper is clearly written.

Some questions
how to compute the integration is (9)? Is it analytically tractable?
typos
line 215: “need to calculated”
line 221: should be reliable estimation
line 242: sampled from


**Time Spent Reviewing:**

6

---

> ### Author Response · Authors · 2021-08-10
> **Response to Reviewer gsQi**
>
> Thank you for your comments and helpful suggestions. Here we provide the corresponding responses to them:
>
> 1. Lack experiments to show the efficiency.
>
> To better show the computation efficiency, we have run an additional experiment with $n$ up to 1000,  in which we separate out the computation of the predictive distributions and directly compare the running time between retraining GPC  (in the naive algorithm) and calculating the joint distribution integral (in the IS+JDI algorithm) when calculating the acquisition function $U^S(x_*)$. The running time with different $n$ is shown in the following table:
>
> |               | n=10| n=100|n=500|n=1000|
> | ----------- | ----------- |------|---|-------|
> | retraining GPC| 0.125	|0.675|64.2|846|
> |  joint distribution integral| 0.568|0.591|0.598|0.643|
>
> It is obvious to see that with increasing $n$, the running time of retraining GPC increases in the cubic order, while the joint distribution integral can be calculated in constant time.
>
> We will also add the original MOCU/SMOCU algorithms into our current query synthesis and pool-based comparison experiments and compare both the error rate and running time among these comparing algorithms.
>
> 2. Ablation studies on the number of samples on more datasets.
>
> With more samples, sampled by both random and IS methods, we can calculate the acquisition function more precisely, which will definitely reduce the error rate. We can expect that as the sample number increases, the error rate of IS will decrease faster than random sampling, since the variance of IS is much smaller. We will verify this tendency on UCI datasets with different number of samples.
>
> 3. How to compute integral (9)?
>
> The integral (9) is not analytically tractable, it is calculated by numerical integration for improper integral (see in Chapter 22 of [b]).
>
> We will carefully proofread our paper and improve the writing with our best efforts.
>
> [b] Chapra, Steven C., and Raymond P. Canale. Numerical methods for engineers. Vol. 1221. New York: Mcgraw-hill, 2011.

---

> > ### Comment · Reviewer_gsQi · 2021-08-30
> > **thanks for your response**
> >
> > thanks to the authors' response. Some more convincing results in terms of computational efficiency is presented. But I'm still not quite convinced to increase my score. Even though I think the contribution is somewhat incremental, I believe this could be a good paper if the experiments are convincing and thorough.

---

> > > ### Author Response · Authors · 2021-08-31
> > > **Response to Reviewer gsQi**
> > >
> > > We thank the reviewer again to reply to our response.
> > >
> > > EER or the more general one-step-look-ahead active learning framework has been known to be computationally inefficient when evaluating acquisition functions. Previous works for pool-based active learning address the computational challenges by approximately retraining the model [3, 6] or subsampling the query pool [11]. We have developed more efficient acquisition function value estimation by our derived joint distribution integral (JDI), which we believe is the first work trying to solve the computation problem without introducing more approximation to evaluate the acquisition function for a general classification model, GPC. Moreover, in addition to pool-based active learning, by avoiding the numerical operations introduced by retraining GPC, we also developed the first gradient-based query synthesis active learning algorithm to the best of our knowledge. We would truly appreciate the reviewer’s reconsideration or detailed reason why there is an impression that our contributions are "incremental".
> > >
> > > Regarding the experiments, we have tried to perform additional experiments based on all the reviewers’ comments. For example, in our response "**Additional performance and running time comparison**", we have shown that our proposed methods, NR-MOCU-RO, NR-SMOCU-RO, and NR-MOCU-SGD,  consistently perform better than state-of-the-art active learning methods, including BALD, EER/MOCU, and SMOCU for computational efficiency as well as sample efficiency (querying less samples to achieve better classification accuracy). If the experiments reported in the original paper and our posted responses were still not convincing enough, we would appreciate it if the reviewer could provide more specific suggestions for additional experiments.

---

### Official Review · Reviewer_ii33 · 2021-07-26

**Rating:** 6
**Confidence:** 3

**Summary:**

This paper proposes to incorporate three ideas in order to improve the efficiency of the classification algorithm based on the Gaussian processes.
(1) Importance sampling to reduce the number of sampling to reliably estimate the acquisition function.
(2) Joint distribution calculation to avoid the retraining of GPC model with a candidate point of query.
(3) Gradient calculation of the acquisition function to efficiently optimize in the query synthesis senario.
Three experiments are conducted to evaluate the proposed approach.
i) With synthetic data to evaluate the approximation error between p (which is estimated by joint distribution calculation) and p' (which is estimated by retraining i.e. naive algorithm).
ii) With synthetic "checkerboard" data to compare the proposed method with conventional approaches:
random, MES and BALD.
iii) With UCI datasets to compare the proposed method with conventional approaches employed in ii) under the pool-based setting.

**Limitations And Societal Impact:**

The authors mention in the main text the limitation of the proposed method that it is difficult to apply it to high dimensional problems because it is based on GCP.
I believe that there is no negative social impact from this research.


**Main Review:**

Originality: This study aims to improve the efficiency of EER-based methods in active learning for gaussian process classification. In this study, three approaches to improve efficiency are proposed, and the proposed method is composed of a combination of them. Among them, the joint distribution calculation to avoid retraining of Gaussian process classification (GCP) model and the gradient computation method to optimize the acquisition function by gradient method in query synthesis senario are considered to be important. The former makes it possible to estimate how much the error will decrease when new data is observed, without the need for computationally expensive retraining of the model. The latter is essential for efficient optimization of the acquisition function by the gradient method in query synthesis scenarios which corresponds to considering a non-countable infinite number of instances.

Quality: This paper technically sounds. How much the proposed method improves the efficiency of GCP-based active learning has been partially evaluated by experiments.
Still, there are a few questions (see Question).

Clarity: The paper is well organized. The explanations of motivation, existing methods and their problems, and the proposed approach are easy to understand.

Significance: While the proposed method provides a general way to improve the efficiency of active learning for GCP, it is especially effective for problems with high data acquisition costs, such as materials science, as mentioned in this paper.

Questions:

- The experiments in section 4.1 show results up to n=100. Is the tendency for the proposed method to be robust (in terms of computation time) to increasing sample size similarly observed for further increases in n?

- Why is MES the only acquisition function for existing methods in the experiments in sections 4.2 and 4.3? I think it is desirable that the comparisons with the more widely used EI and/or UCB (with the error function as the objective function) are conducted.

- The query synthesis scenario considered in the experiment is quite low-dimensional problem (up to two-dimensional). Can we expect the same behavior in higher dimensional spaces?

- SMOCU contains a parameter k that contributes to the approximation to MOCU, how do you adjust this?

**Time Spent Reviewing:**

10h

---

> ### Author Response · Authors · 2021-08-10
> **Response to Reviewer ii33**
>
> We thank the reviewer for the comments. Here we provide the corresponding responses to them:
>
> 1. Running time tendency with increasing $n$:
>
> The tendency would be the same as we increase $n$. To better show the tendency, we have run an additional experiment with $n$ up to 1000, in which we separate out the computation of the predictive distributions and directly compare the running time between retraining GPC (in the naive algorithm) and calculating the joint distribution integral (in the IS+JDI algorithm) when calculating the acquisition function $U^S(x_*)$. The running time with different $n$ is shown in the following table:
>
> |               | n=10| n=100|n = 500|n=1000|
> | ----------- | ----------- |------|---|-------|
> | retraining GPC| 0.125	|0.675|64.2|846|
> |  joint distribution integral| 0.568|0.591|0.598|0.643|
>
> With increasing $n$, the running time of retraining GPC increases in a cubic manner as expected; on the other hand, the joint distribution integral can be calculated in constant time as clearly shown above. We will add these results with discussions and revise Table 1 in our final paper accordingly.
>
> 2. More benchmark algorithms for comparison.
>
> In sections 4.2 and 4.3, we took both MES and BALD as the benchmark algorithms for comparing with the existing algorithms. BALD is among the state-of-the-art methods [5] and MES (uncertainty sampling) is the simplest and most commonly adopted active learning method. We will add more benchmark algorithms into our comparison experiments, including original MOCU/SMOCU and the Approximated Error Reduction (AER) algorithm [3]. Due to the limited time to post the rebuttal, we will report the numbers during the discussion period and add them into the final paper.
>
> EI and UCB are popular acquisition functions for Bayesian Optimization (BO) problems. However, BO assumes the objective function does not change during the learning procedure, while in active learning the error function changes in each iteration as more training data is included (with the updated model posterior), so it is not straight forward to apply EI and UCB into Bayesian active learning.
>
> 3. Performance of query synthesis scenario with higher dimensional space.
>
> We expect that our proposed algorithms perform well in higher dimensional spaces.  In the pool-based active learning experiments we have shown the better performance of our proposed algorithms up to 30 dimensional problems. While we do not include NR-SMOCU-SGD in the pool-based experiments, we can expect SGD still performs well in 30-dimensional space.
>
> 4. Choice of $k$
>
> The optimal choice of $k$ can be affected by the total query budget and the specific classification problem. Generally, $k$ should be small if the sampling budget is large and/or the observation error rate is small; otherwise $k$ should be large. In our experiments, we set $k = 20$ for all the experiments.

---

> > ### Comment · Reviewer_ii33 · 2021-09-02
> > **thanks to the authors' response**
> >
> > Thank you for responding to the review comments and presenting additional experimental results. The new experimental results suggest that the proposed methods, NR-MOCU-RO, NR-SMOCU-RO and NR-SMOCU-SGD, are indeed superior to the baseline method in terms of accuracy. On the other hand, existing methods, especially MES and BALD, are quite good in terms of sample efficiency. When we integrate them and see how much the accuracy is improved for the required size of n, we may not be able to say that the proposed method is completely better than the existing method in this numerical example. Since I believe in the usefulness and prospects of this paper, I will leave my score unchanged.
> >
> > While I understand that it is difficult to convince all readers based on experimental results alone because of the large variability of results depending on the problem in Bayesian active learning , it would have been better to see more examples of applications to real problems.

---

> > > ### Author Response · Authors · 2021-09-02
> > > **Response to Reviewer ii33**
> > >
> > > We thank the reviewer for the reply. We would like to emphasize that we consider sample efficiency as the most important criterion for active learning. In particular in real-world problems, such as materials science applications, obtaining labelled samples can be either difficult or expensive (cost and time). In these cases reducing the number of active learning iterations is more important. For example, in our new numerical experiments for the checkboard function, to get the same accuracy as active learning using NR-MOCU-SGD with 20 new samples, we may need 30 and 27 samples for active learning using MES and BALD, respectively.

---

### Author Response · Authors · 2021-08-29
**Additional performance and running time comparison**

In response to more comprehensive comparisons to existing methods with additional running time comparison, we have finally (we have to program some of the other competing methods ourselves for comparison) obtained additional comparison results on the checkboard $4 \times 4$ problem with the flip error rate = 0.2, comparing both the classification performance and running time.

Three more benchmark methods are included in the experiments. The first two are original MOCU (OR-MOCU) and SMOCU (OR-SMOCU) methods, with their pseudo code provided in Algorithm 1 in the appendix.

The third method ADF-MOCU is based on an approximate inference technique called Assumed Density Filtering (ADF) [6]. ADF only performs moment matching for the new query to update the surrogate GPC model, in contrast to EP that performs iterative moment matching for all labeled data. ADF is faster but less accurate than EP, while the joint distribution integral (JDI) used in our proposed methods has almost the same accuracy as EP (shown in Sec. 4.1 in the original paper). The authors of [6] suggested to use ADF to approximately retrain the model when calculating the acquisition function values for each potential query. Therefore, ADF-MOCU leads to worse sample efficiency than our proposed methods, due to its lower accuracy when computing the acquisition function values. In our implementation of ADF-MOCU, we used ADF to retrain the GPC, and still used importance sampling (IS) to calculate the expectation.

The additional performance comparison results are shown below:

**Error regret:**


 |               | n=10| n=20|n=40|n = 60|
| ----------- | ----------- |------|---|-------|
|random|$0.2003\pm0.0019$|$0.1949\pm0.0019$|$0.1832\pm0.0019$|$0.1724\pm0.0018$|
|MES|$0.1998\pm0.0021$|$0.1899\pm0.0021$|$0.1751\pm0.0022$|$0.1641\pm0.0022$|
|BALD|$0.1965\pm0.0019$|$0.1864\pm0.0020$|$0.1726\pm0.0020$|$0.1609\pm0.0020$|
|OR-MOCU|$0.1943\pm0.0021$|$0.1844\pm0.0021$|$0.1702\pm0.0021$|$0.1595\pm0.0021$|
|OR-SMOCU|$0.1954\pm0.0020$|$0.1862\pm0.0020$|$0.1707\pm0.0020$|$0.1591\pm0.0020$|
|ADF-MOCU|$0.1956\pm0.0020$|$0.1857\pm0.0020$|$0.1703\pm0.0019$|$0.1582\pm0.0020$|
|NR-MOCU-RO|$0.1934\pm0.0020$|$0.1842\pm0.0020$|$0.1682\pm0.0020$|$0.1566\pm0.0021$|
|NR-SMOCU-RO|$0.1929\pm0.0019$|$0.1822\pm0.0019$|$0.1668\pm0.0019$|$0.1559\pm0.0019$|
|NR-SMOCU-SGD|$0.1913\pm0.0017$|$0.1810\pm0.0018$|$0.1671\pm0.0017$|$0.1552\pm0.0017$|

**Running time for each iteration (in seconds):**

 |               | n=10| n=20|n = 40|n=60|
| ----------- | ----------- |------|---|-------|
|random|$0.0\pm0.0$|$0.0\pm0.0$|$0.0\pm0.0$|$0.0\pm0.0$|
|MES|$0.7245\pm0.02$|$0.7174\pm0.02$|$0.7174\pm0.02$|$0.7243\pm0.02$|
|BALD|$1.384\pm0.04$|$1.384\pm0.04$|$1.346\pm0.04$|$1.322\pm0.03$|
|OR-MOCU|$57.29\pm1.60$|$64.47\pm1.85$|$78.07\pm2.01$|$116.2\pm3.19$|
|OR-SMOCU|$60.49\pm1.61$|$65.47\pm1.68$|$84.14\pm2.33$|$119.8\pm3.22$|
|ADF-MOCU|$16.8\pm0.45$|$17.9\pm0.49$|$20.5\pm0.59$|$25.27\pm0.72$|
|NR-MOCU-RO|$31.8\pm1.22$|$33.36\pm1.33$|$35.83\pm1.48$|$43.07\pm2.23$|
|NR-SMOCU-RO|$32.96\pm1.02$|$34.27\pm1.20$|$39.31\pm1.68$|$46.38\pm2.33$|
|NR-SMOCU-SGD|$31.84\pm1.08$|$33.4\pm1.23$|$35.81\pm1.54$|$43.21\pm2.46$|

Based on these new results, we can see clearly that OR-MOCU and OR-SMOCU perform worse than our proposed NR-MOCU-RO and NR-SMOCU-RO for both computation and sample efficiency. That's because when compared with IS, the naive Monte Carlo used in OR-MOCU/OR-SMOCU with GPC cannot accurately estimate the acquisition function values. On the other hand, the running time of OR-MOCU/OR-SMOCU increases faster than NR-MOCU-RO/NR-SMOCU-RO since they retrain the model for every query samples, which requires $O(M_2 n^3)$ computation complexity. ADF-MOCU runs faster than our proposed algorithms, but it performs poorly as the ADF approximation is not accurate enough. Our NR-SMOCU-SGD again performs better than the competing algorithms as it utilizes the gradient information in the optimization procedure. We will add these additional results and provide more details of the experiments.

---

### Author Response · Authors · 2021-09-01
**Any further comments?**

We appreciate the reviewer's comments. We will include all the baseline results in our experiments with different datasets. Is there anything still requiring clarification or additional experiments for the reviewer to reconsider the score?

---

### Decision · Program_Chairs · 2021-09-28

**Decision:**

Accept (Poster)

**Comment:**

The reviewers generally liked the novel ideas presented in the paper about speeding up active learning. Concerns expressed originally in the reviews were mitigated by the rebuttal and a good discussion between the reviewers.
The contributions made in the paper are sufficiently strong to merit publication at NeurIPS. The authors should however make sure to clearly incorporate that answers the questions of the reviewers in the final version.

**Consistency Experiment:**

NeurIPS has a long history of experimentation. In 2014, NeurIPS ran an experiment in which 10% of submissions were reviewed by two independent committees to quantify the randomness in the review process. This year, we repeated a variant of this experiment to see how the quality of the review process has changed over time.  This paper was part of the experiment and was therefore assigned to two committees (consisting of reviewers, an Area Chair, and a Senior Area Chair) that reached independent decisions.  If both committees made the same recommendation, this recommendation was followed. If a single committee recommended acceptance, the paper was accepted (with the exception of a few cases in which the other committee identified what we considered a fatal flaw, e.g., an error in a key result).

This copy’s committee reached the following decision: **Accept (Poster)**

The other committee assigned to the paper recommended **Reject**.  You can find the other set of reviews, along with any follow up discussion with the authors here:
https://openreview.net/forum?id=UK15Hj9qX6I